# Strengthening Resilient Built Environments through Human Social Capital: A Path to Post-COVID-19 Recovery

Oluwagbemiga Paul Agboola [1,*], Hourakhsh Ahmad Nia [2] and Yakubu Aminu Dodo [3]

1 Department of Architecture, Faculty of Engineering and Architecture, Istanbul Gelisim University, Istanbul 34310, Turkey
2 Department of Architecture, Faculty of Engineering and Natural Sciences, Alanya University, Alanya 07400, Turkey; hourakhsh.ahmadnia@alanyauniversity.edu.tr
3 Architectural Engineering Department, College of Engineering, Najran University, Najran 66426, Saudi Arabia; yadodo@nu.edu.sa
* Correspondence: opagboola@gelisim.edu.tr

**Abstract:** There are strong indications that the built environment has had a great influence on the course of the COVID-19 pandemic and the post-disaster recovery. The COVID-19 pandemic has adversely affected both human and global development, while efforts to combat this menace call for an integrated human social capital index. This research seeks to enhance understanding of how the built environment can be enhanced through resilience against the backdrop of the COVID-19 pandemic. This study aims to investigate the impact of a resilient built environment on increasing resilience in the aftermath of the COVID-19 pandemic in Nigeria. The quantitative studies test the impact of four built environment resilience indices (built environment capital, disaster management indices, awareness of the COVID-19 pandemic, and built environment adaptive strategies) on human social capital and COVID-19 pandemic indices. This study reveals the role of human social capital in achieving a resilient built environment in the wake of the COVID-19 pandemic in Nigeria. Built environment capital, disaster management indices, and awareness of COVID-19 also indirectly affect the COVID-19 pandemic indices through human social capital. This study's implications are useful for post-COVID-19 recovery, which is important for future planning of the built environment in Nigeria.

**Keywords:** human social capital; built environment; COVID-19 pandemic; disaster management; structural equation modelling; Nigeria





## 1. Introduction

The COVID-19 pandemic that emerged in 2019 has impacted current people's health status, the economy, and the general growth of the built environment. The pandemic has been the worst for decades; addressing its impacts from the built environment viewpoint is important for driving urban regeneration. Consequently, researchers worldwide are attempting to find the most constructive and productive means of managing the pandemic and limiting its negative impacts [1–3]. The relationship between human interaction, built environments, and the COVID-19 pandemic is paramount to this ongoing debate [2,4].

The built environment, according to [2,4], refers to the human-made physical surroundings in which people live, work, and interact. It encompasses all the structures, spaces, and systems humans create, such as buildings, roads, parks, transportation networks, utilities, and other infrastructure elements. The built environment, which results from human planning, design, construction, and development, plays a significant role in shaping the quality of life, social interactions, and overall well-being of individuals and communities. In view of the COVID-19 pandemic, resilience is an important tool for evaluating urban ecosystems' capacity to adjust to shifting circumstances and meet environmental targets. The notion of resilience has garnered attention across diverse fields since the early 1900s, yet its exploration within the built environment remains limited [5,6].

Within the context of this research, resilience in the built environment is construed in terms of the capacity of a neighbourhood to recover from stresses and pressures while simultaneously fostering constructive adaptation and evolution toward sustainability. The theoretical idea of built-environmental resilience has grown in popularity in recent years, owing to the growing frequency and severity of global disasters [6–8]. This has resulted in the need for a more thorough and holistic method for comprehending the numerous aspects that contribute to built-environmental resilience [9,10].

A handful of studies since the beginning of the COVID-19 pandemic have documented the significance of resilience in the built environment and the relevance of human social capital in recovering from catastrophes [11–13]. These studies describe built-environmental resilience as the capacity of communities and structures to endure, recover, and adapt to numerous hazards such as natural catastrophes, climate change, socioeconomic disturbances, and COVID-19 [14–16]. Other urban hazards that call for resilience include the increasing urban population, which has adversely affected both the environment and the residents of cities [6,14,17]. However, the global COVID-19 pandemic is currently the greatest challenge [18,19].

Human social capital encompasses the assets and values that emerge from interactions between people and interactions with the built environment. Its function in building resilience has attracted a great deal of interest in recent years [9,19,20]. Resilience is crucial for a successful community response to the problems posed by the COVID-19 pandemic. The significance of COVID-19's effect on the built environment cannot be overemphasised. The COVID-19 pandemic has broadened the human horizon by altering people's behaviour and use of the built environment [21,22]. The link between the built environment and the COVID-19 pandemic must receive appropriate attention to foster the potential adaptive construction of cities.

Despite increased awareness of social capital's relevance for resilience, little is known about how it may be quantified and integrated into the built environment. This research tries to fill the vacuum by comprehensively assessing the existing comprehension of the interrelationships between social capital and resilience in the urban landscape. In Nigeria, little research has focused on the diverse interactions between the physical environment and human social capital [23,24]. As a result, this study creates a framework for documenting a resilient built environment using human social capital in the wake of the COVID-19 pandemic.

This research aims to examine the impact of the resilient built environment in response to the COVID-19 pandemic, with a specific focus on the southwest geopolitical zone of Nigeria. The objectives include the following:

(i)    To assess the impact of built environment capital on human social capital during the COVID-19 pandemic.
(ii)   To evaluate the relationship between disaster management indices and human social capital within the framework of a resilient built environment.
(iii)  To explore the degree of people's understanding of the COVID-19 pandemic and its connection to human social capital and the built environment.
(iv)   To investigate the effectiveness of the built environment's adaptive strategies in mitigating the impact of the COVID-19 pandemic on human social capital.
(v)    To analyse the indirect effects of built environment capital, disaster management indices, and COVID-19 awareness on COVID-19 pandemic indices through their influence on human social capital.

This investigation aims to enhance existing knowledge by presenting a methodology that considers the pandemic's indirect effects on built environment capital, disaster management indices, and COVID-19 pandemic awareness via human social capital. Thus, the research proposes a framework to examine the impact of the resilient built environment in the wake of the COVID-19 pandemic, with human social capital being used as a problem-solving technique [9,25,26]. The investigation's results have significant implications for future built-environment development in Nigeria and can help post-COVID-19 recovery

efforts. This study advances the body of knowledge on the significance of human social capital in resilience building and provides policymakers and practitioners in Nigeria and beyond with practical insights.

This article has been structured into the following: Section 2 reviews crucial literature concerning post-COVID-19 pandemic recovery, resilience, human social capital, and the built environment. Section 3 outlines the research framework and formulates hypotheses, while Section 4 details the research's data gathering and analysis methods. Section 5 presents and discusses findings, conclusions, research implications, and reflections for future research endeavours.

## 2. Literature Review and Background

### 2.1. Post-COVID-19 Pandemic Recovery

The COVID-19 pandemic caused deaths across the globe [27], and the African Continent was not spared [28], as shown in Figures 1 and 2, respectively. These had enormous consequences for communities and economies, resulting in extensive social, economic, and health implications. Under these circumstances, the significance of social capital in facilitating recovery has grown in importance. Several research efforts have already shown the significance of social capital and cooperation among individuals and groups to overcome challenges [20,25,27]. Some literature substantiates that social capital can help achieve resilience after the pandemic [8,28,29], and facilitate communities' adaptation goals after catastrophes. Social capital has become vital in assisting communities to cope with the pandemic's effects and build resilience. There is a need for coordination and cooperation to assist those most affected by the pandemic's social and economic effects [30–32].

In disaster and emergency research, resilience and vulnerability are increasingly studied alongside social capital [33–35]. Before and after a crisis, social capital influences resilience and vulnerability. It is often used to assess people's potential to recover from disasters [8,36,37]. Simply put, social capital includes the moral codes, principles, confidence, and connections of communities. The term also embraces social organisations in which members of society help one another, thus boosting urban communities and lessening dependence on the state. These organisations may possess analytics and insights for group cohesion and enhancing cooperation and collaboration in the face of disasters.

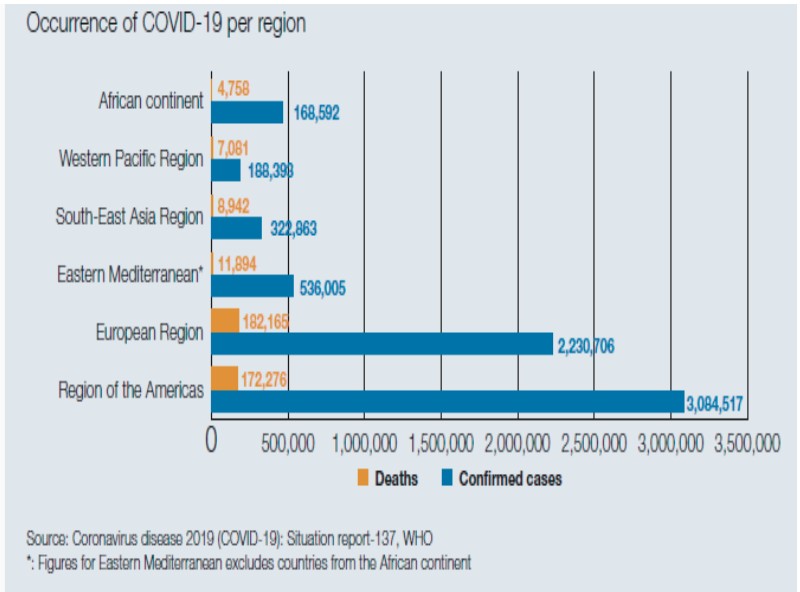

**Figure 1.** COVID-19 pandemic occurrence across the globe. Source: [38].

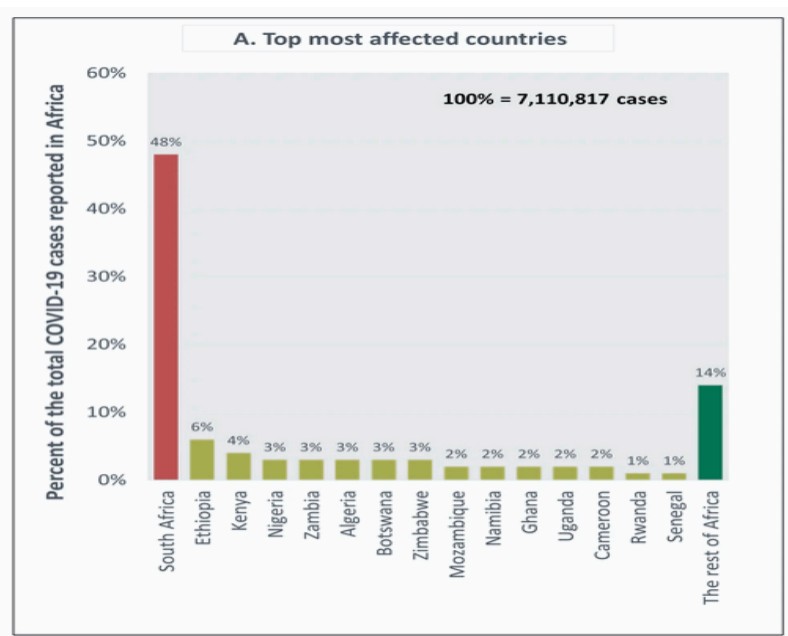

**Figure 2.** The African countries most significantly impacted by COVID-19. Source: [38].

*2.2. Resilience, Human Social Capital, and the Built Environment*

Resilience is described as the capacity to absorb disruptive changes while recognising and capitalising on opportunities [39–41]. In line with this assertion, Ref. [40] introduced the concept of resilience as a series of adaptive abilities that guide positive functioning following a disturbance. These studies inferred that resilience is not only an ultimate result but also an intermediary process in the form of adaptive capacity to achieve desired results. In the context of the built environment, resilience is considered to be a city's capability to endure while recovering from natural disasters, economic shocks, and other disruptive events. Human social capital is the value derived from social networks, relationships, and interactions between people. It is essential in the built environment for building resilient communities after disruptive events. There are several ways in which resilience and human social capital are interconnected in the built environment, namely:

(i)   Building social networks and relationships: Social capital entails forming relationships between individuals and groups. These networks and relationships can be leveraged in times of crisis to provide support and resources to those who need them [37].

(ii)  Promoting community engagement and participation: Resilient communities actively participate in planning and developing their built environment. By promoting community engagement and participation, social capital enhances the community's resilience as a whole [29,42].

(iii) Fostering trust and cooperation: Trust and cooperation are essential for building social capital and promoting resilience in the built environment. When individuals and groups trust each other and cooperate, they will recover better from disruptive events [25].

(iv)  Encouraging knowledge sharing and learning: Resilient communities gain insight from previous difficulties and adapt their approaches to potential challenges. Social capital is essential for promoting knowledge sharing and learning among individuals and groups within the neighbourhood [43].

The interplay between urban resilience, human social capital, and the built environment is presented in Figure 3, as adapted from the study of [25,39,43]. By building strong social networks and relationships, promoting community engagement and participation, fostering trust and cooperation, and encouraging knowledge sharing and learning, communities can become more resilient and adapt to disruptive events.

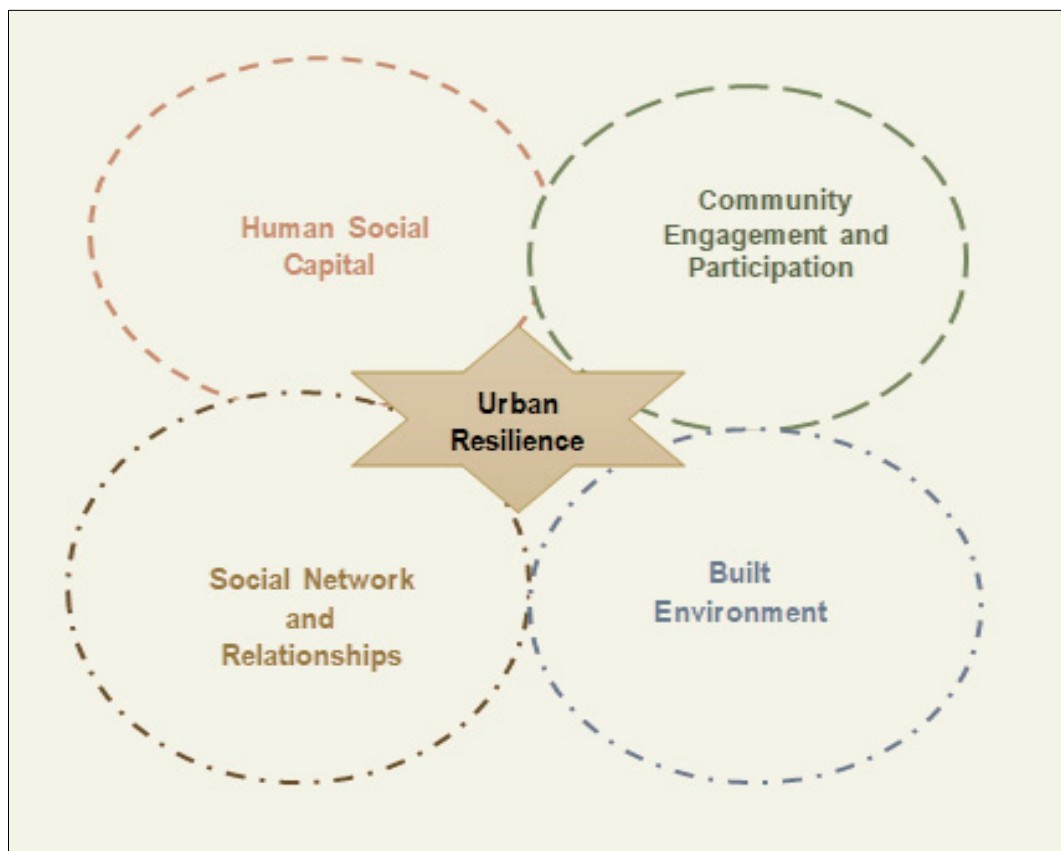

**Figure 3.** Social relationships and urban resilience. Source: [25,39,43].

### 2.3. Consolidating a Resilient Built Environment Using Human Social Capital

The built environment comprises the structural, ecological, and socio-cultural capital represented by man-made buildings and infrastructure. The built environment's resilience is crucial in light of the growing frequency and severity of natural catastrophes and other stressors such as climate change and socioeconomic disturbances. A resilient built environment is characterised by a structure's ability to endure and adapt to various shocks. Accomplishing this level of resilience necessitates the skill and knowledge to consider multiple social, economic, and technical aspects.

The idea of resilience provides a way to manage the protracted adaptation of the built environment and investigate the impacts of ecological changes on the effectiveness of various planning, design, and management approaches. In consequence, resilience is a conceptual and modelling framework that identifies the processes that help or hinder the attainment of sustainable environmental objectives. Resilience has three components: (i) the term's core description, the capacity to withstand or recover from difficulties; (ii) models for translating the ambiguously defined core concept to specific situations; and (iii) an analogy for the social and private assumptions, experiences, and values associated with the theory. Social capital is one major aspect progressively recognised as important for generating resilience in the built environment. Trust and reciprocity rules can increase individuals' desire to cooperate and contribute to the collective good [29,43].

Human social capital is important in fostering resilience because it provides individuals and communities with the resources to cope with adversity. It can improve access to resources, facilitate risk reduction and adaptation, and improve communities' ability to respond to catastrophes [28,37,44]. Social networks can provide emotional support, information, and access to resources to help people overcome challenges and bounce back from setbacks. Strong social connections can also help build trust and cooperation, which are essential for effective collaboration and problem-solving [25]. This entails integrating

social, economic, and technological aspects, as well as addressing inter-generational justice in creating resilience. Despite growing recognition of the value of social capital for resilience, incorporating social capital into the physical environment's design is still in its early stages. The difficulty in measuring and quantifying social capital and the necessity to address power dynamics and inequality within communities are some of the obstacles and limitations of employing social capital as a resilience-building method [32,36,44].

A rising corpus of research on resilience in the built environment has underlined the necessity for a systems approach that takes into account the interconnections between diverse elements. The built environment, which refers to the physical surroundings in which people live and work, also significantly promotes resilience. The design and layout of buildings, neighbourhoods, and cities influence the social interactions and relationships that occur within them. For example, mixed-use developments that combine residential and commercial spaces can promote social interaction and community engagement, while poorly designed neighbourhoods with few public spaces and amenities can result in social exclusion and disengagement [25].

The built environment can also impact resilience due to its ability to adapt to environmental disasters and other disruptions. Resilient infrastructure, such as buildings, roads, and energy systems, can help minimise the consequences of catastrophes and ensure that communities recover more quickly after a crisis. Consequently, the resilience concept relates to both human social capital and the built environment. Building strong social connections and creating resilient infrastructure may assist people and communities in adapting to and recovering from adversity, while poorly designed environments and weak social networks can hinder resilience and exacerbate the impact of stress and trauma.

### 2.4. Conceptual Framework

Protection motivation theory (PMT) was devised to understand people's reactions to fear-based emotional appeals. It proposes that people protect themselves from perceived danger based on a combination of threat and coping appraisal. It is mostly used in the biomedical sciences to urge changes in health habits [45]. Over time, scholars have employed PMT in social, environmental, and psychological research [46]. People will take immediate action if they perceive a significant danger, such as apprehension about getting and transmitting COVID-19, and wish to curtail the spread of the virus. In light of this, this study incorporated 'resilience' and 'human social capital' into PMT as a framework, as shown in Figure 4.

Scholarly research into crises and disasters typically employs the same theories to illustrate and evaluate societal manifestations, such as (i) how communities, organisations, or individuals respond to traumatic emotions; (ii) the socioeconomic or political ramifications of catastrophes; and (iii) what changes in humanity's fabric are needed to mitigate a crisis. On this premise, an interaction between social capital and resilience is affirmed by [47,48]. The two are frequently utilised during pandemics, terrorist attacks, and disasters. Similar studies by [37,48] affirmed the interconnections between resilience and catastrophe management. Catastrophe and disaster research have advanced our understanding of resilience, social capital, and disaster management via scientific, analytical, and investigatory studies.

### 2.5. Hypothesis Development

This study employs multivariate data analysis and confirmatory factor analysis (CFA) by AMOS (Analysis of Moment Structures) to test the proposed framework and research hypotheses. The rationale for selecting AMOS and CFA is that they provide a powerful method for testing the validity of a theoretical model. AMOS is particularly well-suited for conducting CFA because it provides a user-friendly graphical interface that allows researchers to build complex SEM models with ease. It includes a variety of useful features for model estimation, model fit testing, and model modification. CFA is a statistical technique used to test the degree to which a group of observed variables can be accounted for by a lower number of latent factors. In other words, CFA allows researchers to confirm

whether their measures determine what they are supposed to measure and whether they are related to the underlying constructs they are supposed to represent.

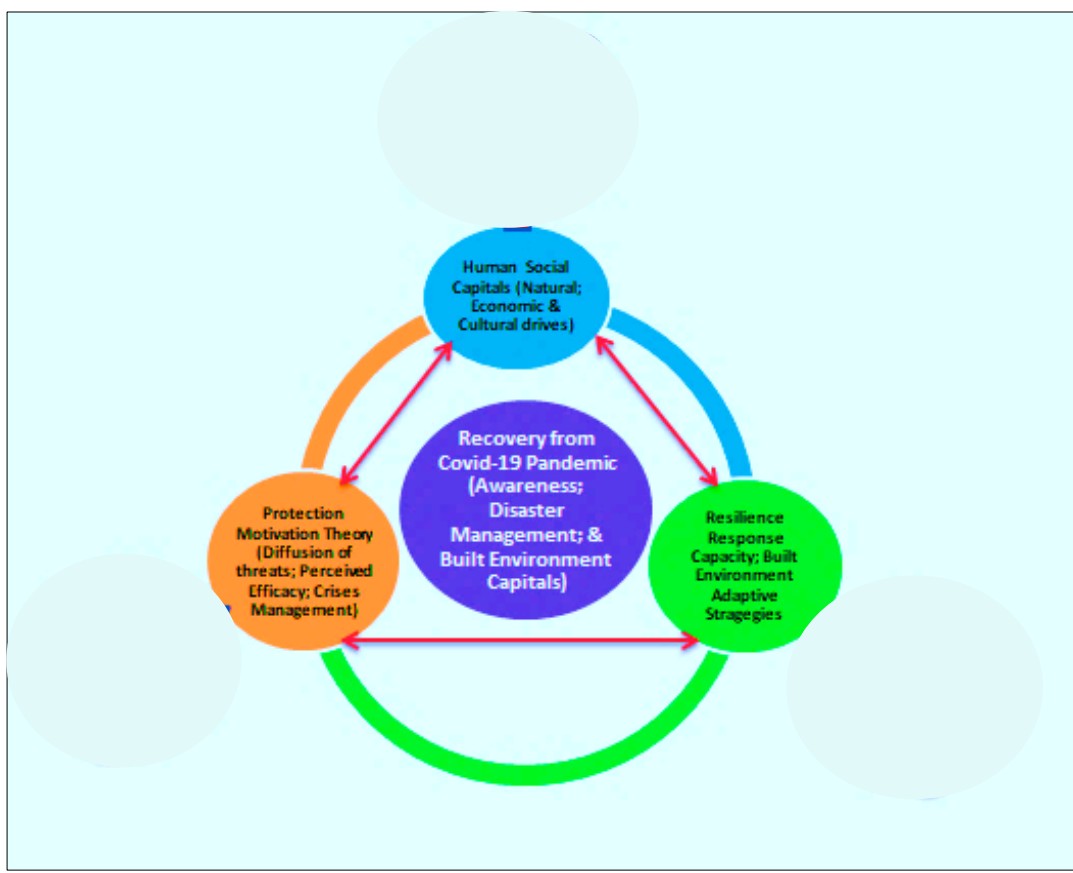

**Figure 4.** Theoretical framework, capturing protection motivation theory, resilience, and human social capital. (Author's Conceptualisation).

This empirical analysis documents the significant benefits of four resilient built environment indicators (built environment capital, disaster management indices, COVID-19 pandemic awareness, and built environment adaptive methods) in ameliorating the adverse effects of the COVID-19 pandemic. Figure 5 depicts a hypothetical analytical regression model of the effects of human social capital on COVID-19 pandemic indicators based on this premise. Built environment capital, disaster management indices, and COVID-19 pandemic awareness indirectly affect COVID-19 pandemic indices via human social capital. Consequently, the following hypotheses are formulated:

- **Hypothesis H1a**: *Enhanced built environments improve COVID-19 pandemic indicators.*
- **Hypothesis H1b**: *Enhanced built environments positively influence human social indicators.*
- **Hypothesis H2a**: *Effective disaster management measures improve COVID-19 pandemic indicators.*
- **Hypothesis H2b**: *Effective disaster management initiatives positively influence human social capital indicators.*
- **Hypothesis H3a**: *Increased awareness of the COVID-19 pandemic improves COVID-19 pandemic indicators.*
- **Hypothesis H3b**: *Increased awareness of the COVID-19 pandemic positively influences human social capital indicators.*
- **Hypothesis H4a**: *Adaptive strategies within built environments improve COVID-19 pandemic indicators.*
- **Hypothesis H4b**: *Adaptive strategies within built environments positively influence human social capital.*

- **Hypothesis H5**: *Human social capital improves COVID-19 pandemic indicators.*

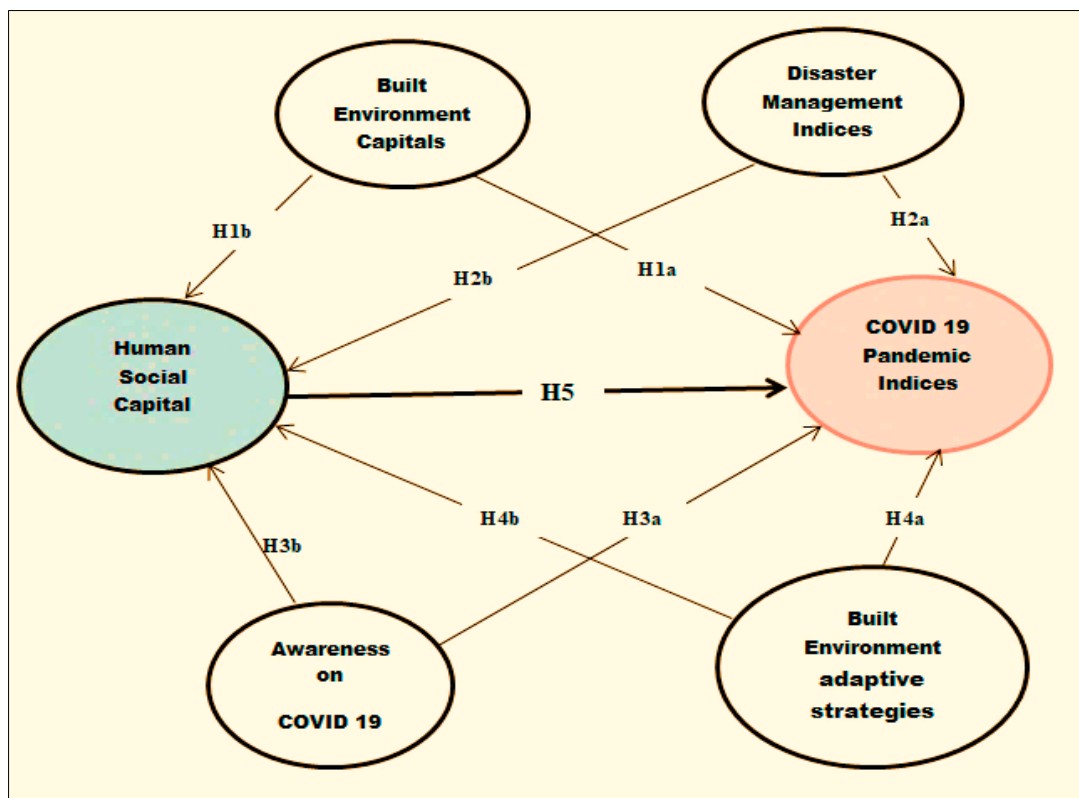

**Figure 5.** Hypothetical analysis and regression model of the impacts of human social capital on COVID-19 pandemic indicators. Author's conceptualisation.

## 3. Research Design and Methods

### 3.1. Variable Measurement

The contents of the survey questionnaire were divided into two segments, as presented in Table 1. The first segment comprises questions about the respondent's demographic profile, such as age, gender, work status, and educational background. The second segment assessed the various constructs included in the proposed framework, such as built environment capital, disaster management indices, COVID-19 pandemic awareness, human social capital, and COVID-19 pandemic indices. The measurement items were all drawn from established scales, and the content of some items was modified to fit the research context.

Validating a research questionnaire is important for ensuring that it accurately measures targeted research questions and produces reliable and meaningful data. The questionnaire's construct (variable) is well-defined and aligns with our research variables. A pilot survey was carried out to ensure the content validity of the questionnaires, in which experts in the field were asked to review the measurement variables and provide. They provided a favourable assessment of the questions. Pilot testing was accomplished with a sample of 25 members of the target population, which helped identify issues with the wording of questions, instructions, or response options. Participants' feedback during the pilot test was used to refine the main questionnaire.

All the items were adjusted to positive questions rated on a five-point Likert scale. Respondents' opinions on the research questions were measured from 1, 'strongly disagree', to 5, 'strongly agree'. The measurement of built environment capital was based on four items adopted from [49,50] in which the literature elucidates the interconnection between natural, human, economic, and cultural capital for a sustainable built environment. Built environment capital (BEC) influences the type and level of services and goods necessary for satisfying social desires. This necessitates the existence of natural capital

and the capability to utilise it through constructed capital, thereby satisfying human wants through commodities and services [49,51]. In the built environment capital section of the questionnaire, the respondents were asked how strongly they agreed with statements such as "*Human capital relates to low-energy resources and services*" and "*Human capital is a production element that interacts with constructed capital to achieve economic consequences*", amongst others.

The disaster management indices (DMI) measurements were based on seven items derived from [52], according to which the purpose of disaster management is to offer a proper reaction to and retroactive effect after a disaster. The effectiveness of this response is determined by the readiness level of both the accountable institutions and the populace in their entirety. The aim is to respond effectively and properly when a threat becomes a disaster. The performance indicators of disaster-relief agencies include effective organisational capabilities and the competence and strategies in place to deal with catastrophic repercussions. The questions included '*Disaster management can be reduced through the organisation and coordination of emergency operations*', and '*Disaster management can be reduced through hazard monitoring and forecasting*', among others.

Awareness of the COVID-19 pandemic (AWC) was measured by six tested items, as previously measured by [53]. Questions include: '*When it comes to COVID-19, I am terrified*', '*When I think about the deaths caused by COVID-19, I get nervous*', and so forth. The COVID-19 pandemic indicators (COVIN) were measured using five previously tested items conforming to [8,53]. A sample of the questions includes: '*I am helpless in the face of the COVID-19 pandemic*', '*During COVID-19, I was concerned that I lacked sufficient immunity to adequately combat the Coronavirus*'. Built-environment adaptive strategies included five items adapted from [14,15,54]. Human social capital included five measurement items adapted from [42,51]. This was with the view of documenting the role played by human social capital in enhancing the resilience of the built environment after the COVID-19 pandemic.

**Table 1.** Questionnaire development and variable measurements.

| Sections | Factors | No. of Items | Scales | Sources |
|---|---|---|---|---|
| | Section One | | | |
| 1. | Demographic | Multiple | Nominal and ordinal | Researchers |
| 2. | Respondents' self-assessed awareness about the built environment | One | Ordinal | Researchers |
| 3. | Respondents' self-assessed awareness about the COVID-19 pandemic | One | Ordinal | Researchers |
| | Section Two | | | |
| 4. | Human social capital | Five measurement items | Likert scale | [42] |
| 5. | Built environment capital | Six tested items | Likert scale | [49,50] |
| 6. | Disaster management indices/indicators | Seven tested items | Likert scale | [52] |
| 7. | COVID-19 pandemic awareness | Six tested items | Likert scale | [53] |
| 8. | COVID-19 pandemic indices/indicators | Five tested items | Likert scale | [53] |

### 3.2. Sampling and Data Analysis

This study used an online poll to gather data from a sample of Nigerian respondents via an emailed Google Forms questionnaire between 2 February and 30 April 2022. In total, 427 acceptable random samples were obtained from the survey, and the demographics

of the respondents accurately reflected those of Nigeria. Table 2 indicates the results of the measurement of sampling adequacy. The scale's reliability and internal consistency were assessed using Cronbach's alpha coefficient test. Overall, the scale was established to be adequate, with KMO values of 0.870. The sample was deemed satisfactory based on the minimum standards of a Cronbach value of 0.700 or higher suggested by [55]. The Cronbach's alpha values of 0.853 and 0.896 obtained in this study confirm its reliability for subsequent analysis.

**Table 2.** Sampling adequacy.

| KMO and Bartlett's Test | | Value |
|---|---|---|
| Measurement of Sampling Adequacy | | 0.870 |
| Sphericity | Chi-square value | 19,616.712 |
| | Df value | 2775 |
| | Sig/supported | 0.000 |

Normality tests were carried out to show the statistical model dataset. The normal distribution set is based on reports of skewness and kurtosis [55]. The skewness and kurtosis Z-values should be greater than 1.96, and the p-value for the Shapiro-Wilk estimate should be greater than 0.05. These were eventually achieved, as 86 percent of the skewed Z-values were greater than 1.96, which was sufficient for further investigation [55,56]. A total of 31 outliers were identified and eliminated.

The data analysis was evaluated in two stages, as recommended by [57,58]. It was carried out via measurement and structural modelling [58]. Developing a measurement model and performing the first synchronised CFA included all variables. The second procedure took into consideration the use of structural modelling to examine the links between the components, as shown in Figure 2. For the mediation effects, comparisons were made between the mediation models, while bootstrapping was used, as suggested by [59], to test for the significance of the indirect effects.

This quantitative survey uses SPSS and AMOS version 24.0 to explore the enhancement of a resilient built environment through the use of residents' social capital in the post-COVID-19 pandemic era. This quantitative research strategy was deemed suitable for mitigating biassed evaluation and discussion, as supported by the works of [60,61]. A sample size of 100 or above was regarded as adequate for variance-based structural equation modelling [55]. The sample size was calculated using the Raosoft sample size calculator [62] by presuming a 95% confidence level, a 5% margin of error, and response distribution of 50%, yielding a sample size of 427.

The screened data obtained were evaluated with multivariate data analysis and confirmatory factor analysis using AMOS software version 24. AMOS (Analysis of Moment Structures) is a popular software tool used for structural equation modelling (SEM), a statistical technique for testing complex relationships between variables. AMOS allows researchers to build models that depict the hypothesised relationships between variables and then test those models using confirmatory factor analysis (CFA). For example, AMOS can compute a variety of goodness-of-fit indices, such as the chi-square statistic, the comparative fit index (CFI), and the root mean square error of approximation (RMSEA), which can help researchers evaluate the overall fit of their model. The confirmatory factor analysis was used to evaluate the reliability and validity of each construct in the model, while multivariate data analysis was used to test the research hypotheses. Before cross-validation, exploratory factor (EF) analysis yielded a valuable model-specific algorithmic technique with confirmatory factor analysis.

The EFA variables were related to the latent construct, whereas the CFA defined the plethora of indicators expected for the outcomes. CFA can be defined as a method for validating or rejecting an estimate [55]. As an exploratory study, it determines the research instrument's validity using an exploratory factor analysis approach. The data collected

were analysed using the proposed framework, and the results were used to demonstrate the indirect effects of built environment capital, disaster management indices, and COVID-19 pandemic awareness on the pandemic through human social capital. The contents of the survey questionnaire allowed feedback from the respondents on the impact of a resilient built environment on post-COVID-19 recovery in Nigeria.

## 4. Results

The purpose of this investigation is to test the research hypotheses by analysing the data gathered on people's opinions. Five hundred questionnaires were sent out, and 438 were returned. The eventual 427 valid samples amounted to an 85.4% valid response rate, guaranteeing statistical validity with a 95% confidence interval and a ±0.05 sampling error. This research data are adequate based on the targeted population, as suggested by [62]. This sample size ensures that this study's results are reliable, statistically meaningful, and can be generalised to a larger population.

Demographic statistics revealed that 57.37% of respondents were male, 42.62% were female, 13.80% were 18–35 years old, 37.00% had a Bachelor's/HND degree, and 22.48% worked in healthcare services. This study's sample size ensures a representative cross-section of the population under investigation. Also, this sample adequately reflects the diversity and characteristics of the larger population to draw meaningful conclusions that allow generalisation. Table 3 shows the demographic features.

**Table 3.** Participant profiles (*N* = 427).

| Factors | Categorisation | Frequency (N) | Percentage (%) |
|---|---|---|---|
| Gender | Male | 245 | 57.37 |
| | Female | 182 | 42.62 |
| Age | 18–35 | 59 | 13.80 |
| | 36–45 | 124 | 29.03 |
| | 46–50 | 162 | 37.93 |
| | 51 and above | 82 | 19.20 |
| Work Status | Researchers/lecturers | 80 | 18.73 |
| | Construction workers | 61 | 14.28 |
| | Healthcare service workers | 96 | 22.48 |
| | Public service workers | 92 | 21.54 |
| | Industrial workers | 98 | 22.95 |
| Education | OND/NCE | 109 | 25.52 |
| | Bachelor's degree (BSc.)/HND | 158 | 37.00 |
| | Master's degree (MSc.) | 102 | 23.88 |
| | Doctoral degree (Ph.D.) | 58 | 13.58 |
| Rate your knowledge about the built environment. | Excellent | 186 | 43.55 |
| | Good | 128 | 29.97 |
| | Moderate | 98 | 22.95 |
| | Very poor | 15 | 3.50 |

**Table 3.** *Cont.*

| Factors | Categorisation | Frequency (N) | Percentage (%) |
|---|---|---|---|
| Rate your knowledge about COVID-19. | Excellent | 160 | 37.47 |
| | Good | 196 | 45.90 |
| | Moderate | 58 | 13.58 |
| | Very poor | 13 | 3.04 |
| Rate your knowledge about COVID-19. | Excellent | 160 | 37.47 |
| | Good | 196 | 45.90 |
| | Moderate | 58 | 13.58 |
| | Very poor | 13 | 3.04 |

Figures 6 and 7 indicate respondents' self-ratings of their knowledge regarding the built environment and the pandemic. The multivariate data analysis method was applied to examine and test the research hypotheses. The measurement model consisted of six latent constructs, namely built environment capital (four items), disaster management indices (seven items), awareness of the COVID-19 pandemic (six items), COVID-19 pandemic indices (five items), human social capital (five items), and built environment adaptive strategies (five items). The principal component analysis (PCA) identified and categorised variables into major components; the results are presented in Table 4. The assessment of 32 variables came up with six major components that significantly converged after six iterations. The iterations explain 92.72% of the overall variance, with variance percentages of 15.06%, 15.14%, 15.50%, 15.13%, 15.11%, and 15.78%, respectively. These percentages demonstrate that a combination of these six components had substantial percentages (92.72%) of the original data's variability.

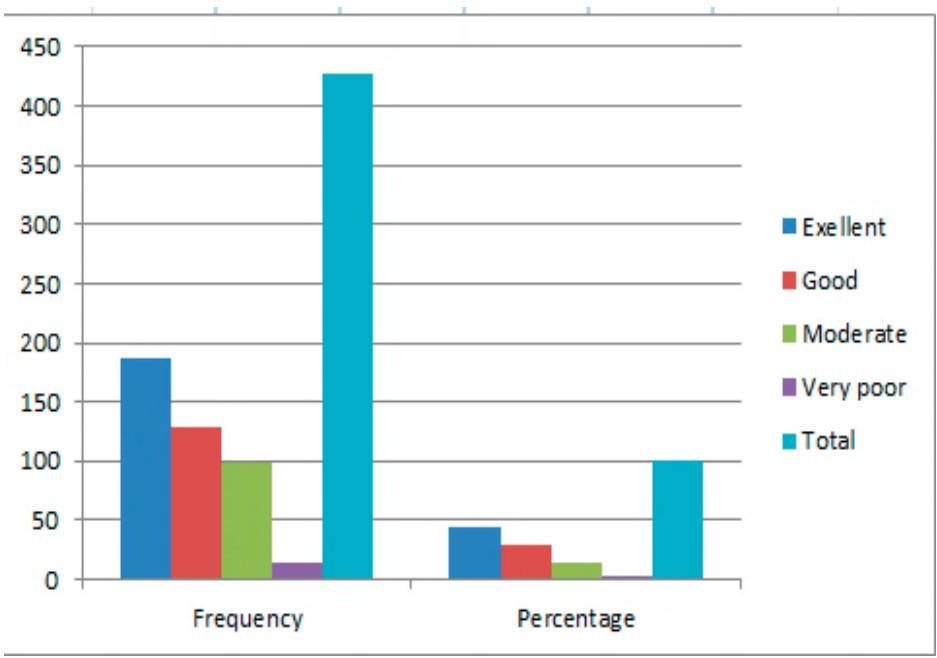

**Figure 6.** Respondents' self-assessed awareness about the built environment.

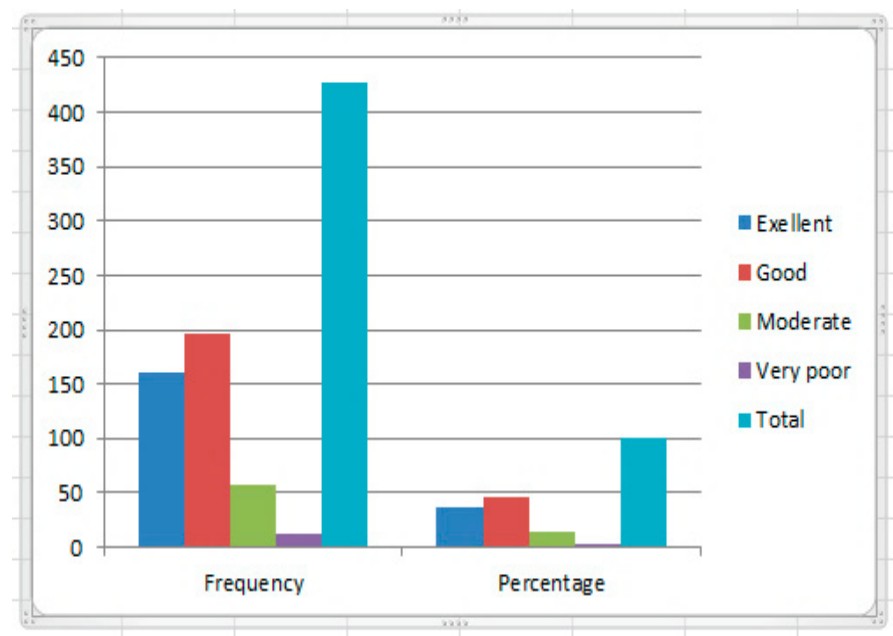

**Figure 7.** Respondents' self-assessed awareness about the COVID-19 pandemic.

**Table 4.** The extracted principal component analysis for the constructs.

| Variables | Components | | | | | |
|---|---|---|---|---|---|---|
| | 1(BEC) | 2(DMI) | 3(AWC) | 4(COVIN) | 5(BEAS) | 6(HSC) |
| BEC1 | 0.821 | | | | | |
| BEC2 | 0.799 | | | | | |
| BEC3 | 0.773 | | | | | |
| BEC4 | 0.725 | | | | | |
| DMI 1 | | 0.859 | | | | |
| DMI 2 | | 0.842 | | | | |
| DMI 3 | | 0.705 | | | | |
| DMI 4 | | 0.814 | | | | |
| DMI 5 | | 0.824 | | | | |
| DMI 6 | | 0.742 | | | | |
| DMI 7 | | 0.814 | | | | |
| AWC 1 | | | 0.828 | | | |
| AWC 2 | | | 0.887 | | | |
| AWC 3 | | | 0.846 | | | |
| AWC 4 | | | 0.847 | | | |
| AWC 5 | | | 0.834 | | | |
| AWC 6 | | | 0.772 | | | |
| COVIN 1 | | | | 0.859 | | |
| COVIN 2 | | | | 0.855 | | |
| COVIN 3 | | | | 0.782 | | |
| COVIN 4 | | | | 0.796 | | |
| COVIN 5 | | | | 0.847 | | |

**Table 4.** *Cont.*

| Variables | Components | | | | | |
|---|---|---|---|---|---|---|
| BEAS1 | | | | | 0.791 | |
| BEAS2 | | | | | 0.789 | |
| BEAS3 | | | | | 0.798 | |
| BEAS4 | | | | | 0.758 | |
| BEAS5 | | | | | 0.781 | |
| HSC1 | | | | | | 0.857 |
| HSC2 | | | | | | 0.841 |
| HSC3 | | | | | | 0.874 |
| HSC4 | | | | | | 0.861 |
| HSC5 | | | | | | 0.854 |
| %Variance explained | 15.06% | 15.14% | 15.50% | 15.13% | 15.11% | 15.78% |

SPSS version 22.0 was used for reliability analyses, while AMOS version 24.0 was used for a validity test using the CFA results. We also conducted item reliability and a convergent validity study for each construct in the model in addition to the overall assessment of model fit; Table 5 presents the results. The measurement model fit has Chi-square/df smaller than 3, CFI, TLI, and IFI are all larger than 0.9, and RMSEA is smaller than 0.08, in line with the suggestions of [55,63].

**Table 5.** The measurement constructs show reliability and convergent validity.

| Constructs/Variables | Item Codes | Standard Loadings | Cronbach's Alpha ($\alpha$ > 0.7) | Composite Reliability (CR > 0.7) | AVE (AVE > 0.5) |
|---|---|---|---|---|---|
| **Built Environment Capital (BEC)** | | | **0.853** | **0.876** | **0.78** |
| Built environment capital relates to low-energy resources. | BEC1 | 0.821 | | | |
| Built environment capital could achieve economic consequences. | BEC2 | 0.799 | | | |
| Built environment capital is sufficient for human satisfaction. | BEC3 | 0.773 | | | |
| Built environment capital could improve cultural sustainability. | BEC4 | 0.725 | | | |
| **Disaster Management Indices (DMI)** | **Items Codes** | | **0.879** | **0.847** | **0.87** |
| Disaster management reduces emergency operations. | DMI 1 | 0.859 | | | |
| Disaster management can be reduced through forecasting. | DMI 2 | 0.842 | | | |
| Disaster management can be reduced through hazard evaluation. | DMI 3 | 0.705 | | | |

| Constructs/Variables | Item Codes | Standard Loadings | Cronbach's Alpha ($\alpha > 0.7$) | Composite Reliability (CR > 0.7) | AVE (AVE > 0.5) |
|---|---|---|---|---|---|
| Disaster management can be reduced through risk assessment. | DMI 4 | 0.814 | | | |
| Disaster management can be reduced through community participation. | DMI 5 | 0.824 | | | |
| Disaster management can be reduced through training and education. | DMI 6 | 0.742 | | | |
| Disaster management can be reduced through protection techniques. | DMI 7 | 0.814 | | | |
| **Awareness of the COVID-19 Pandemic (AWC)** | **Item codes** | | 0.883 | 0.879 | 0.78 |
| When it comes to the COVID-19 pandemic, I am terrified. | AWC 1 | 0.828 | | | |
| When I think about the deaths caused by the COVID-19 pandemic, I get nervous. | AWC 2 | 0.887 | | | |
| I am terrified of contracting COVID-19. | AWC 3 | 0.846 | | | |
| I am frightened of dying as a result of the COVID-19 pandemic. | AWC 4 | 0.847 | | | |
| When I heard about the number of deaths caused by the COVID-19 pandemic, I felt afraid and sad. | AWC 5 | 0.834 | | | |
| Regarding the COVID-19 pandemic, I am concerned about the future. | AWC 6 | 0.772 | | | |
| **COVID-19 Pandemic Indicators (COVIN)** | **Items Codes** | | 0.875 | 0.825 | 0.74 |
| I am helpless in the face of the COVID-19 pandemic. | COVIN 1 | 0.859 | | | |
| I am restless in the face of the COVID-19 pandemic. | COVIN 2 | 0.855 | | | |
| I felt the sensation of control throughout the COVID-19 pandemic. | COVIN 3 | 0.782 | | | |
| During the COVID-19 pandemic, I was concerned that I lacked sufficient immunity. | COVIN 4 | 0.796 | | | |
| When I think of the COVID-19 pandemic, I think of how precious life is. | COVIN 5 | 0.847 | | | |

**Table 5.** *Cont.*

| Constructs/Variables | Item Codes | Standard Loadings | Cronbach's Alpha (α > 0.7) | Composite Reliability (CR > 0.7) | AVE (AVE > 0.5) |
|---|---|---|---|---|---|
| **Built Environment Adaptive Strategies (BEA)** | **Item codes** | | **0.858** | **0.877** | **0.75** |
| Built environment adaptive strategies involve the use of sustainable landscaping methods. | BEAS1 | 0.791 | | | |
| Built environment adaptive strategies involve the use of technology. | BEAS2 | 0.789 | | | |
| Sustainable urban drainage systems improve built environment adaptation strategies. | BEAS3 | 0.798 | | | |
| Built environment adaptive strategies involve using new technology. | BEAS4 | 0.758 | | | |
| Built environment adaptive strategies involve supporting public environmental awareness. | BEAS5 | 0.781 | | | |
| **Human Social Capital (HSC)** | **Items Codes** | | **0.896** | **0.901** | **0.78** |
| Human social capital involves positive stakeholder participation. | HSC1 | 0.857 | | | |
| Human social capital involves human ecological work connections. | HSC2 | 0.841 | | | |
| Human social capital involves residents' neighbourhood connections. | HSC3 | 0.874 | | | |
| Human social capital involves human activities that could decrease vulnerability. | HSC4 | 0.861 | | | |
| Human social capital involves feelings of trust and safety. | HSC5 | 0.854 | | | |

- Cronbach's alpha was used to determine structural dependability or internal consistency, which ranged from 0.853 to 0.896 for all constructs, above [64]'s criteria of 0.7. As a result, the scales for all structures are reliable. Overall, for the model fit of the measurement model, the analysis achieved sufficient measures ($\chi2/df$ = 3.206 and RMSEA = 0.052, CFI = 0.939, GFI = 0.947, TLI = 0.918, and IFI = 0.937). Each construct's standardised item loadings were statistically significant ($p < 0.001$). None of the items had loadings smaller than 0.50, a common factor analysis threshold [65,66]. The composite reliability [66] and the average variance retrieved [67] were also examined for each construct. The composite dependability of a concept evaluates its unidimensionality and should, at the very least, be greater than the 0.70 cut-off standard [68]. These criteria are met by all of our structures.

The extracted average variance estimates the proportion of variation attributable to random error [69]. All of our measures are greater than 0.50, indicating that we have strong

internal consistency and that the variance captured by each construct is greater than the variances caused by measurement error [67]. These findings indicate that the six latent components have appropriate convergent validity and item reliability. The degree to which one latent construct differs from another is referred to as discriminant validity. The AVE was used to confirm discriminant validity in this current study. This was achieved by coordinating the correlations among the latent constructs with the square roots of the retrieved average variance. The square root of the extracted average variance exceeded the correlations among latent components, indicating good discriminant validity, as shown in Table 6.

**Table 6.** Discriminant validity.

| Construct Variables | HSC | BEC | DMI | AWC | BEAS | COVIN |
|---|---|---|---|---|---|---|
| Built Environment Capital | 0.861 | 0.020 | 0.030 | 0.120 | 0.060 | 0.033 |
| Disaster management indices | 0.644 | **0.822** | 0.021 | 0.062 | 0.030 | 0.032 |
| Awareness of COVID-19 pandemic | 0.611 | 0.751 | **0.868** | 0.010 | | |
| Built environment adaptive strategies | 0.682 | 0.622 | 0.735 | **0.820** | 0.030 | 0.010 |
| Human social capital | 0.551 | 0.561 | 0.557 | 0.734 | **0.897** | 0.030 |
| COVID-19 pandemic indices | 0.675 | 0.613 | 0.638 | 0.662 | 0.734 | **0.876** |

**Note**: Bold values face represents the square root of the average variance extracted. The average variance extracted (AVE) values were greater than the square of the correlation estimates between the constructs. ***BEC**—built environment capital*; ***DMI**—disaster management indices*; ***AWC**—awareness of* the *COVID-19 pandemic*; ***BEAS**—built environment adaptive strategies*; ***COVIN**—COVID-19 pandemic indices*; ***HSC**—human social capital*.

Principal component analysis was used to extract the data in six iterations, and the rotation converged.

The results of the analysis of the model's structural component are shown in Figure 8. The control variables are not shown in this diagram for clarity purposes. Nonetheless, each dependent construct was represented in the structural model as previously suggested [70]. The results of the final structural model achieved a good overall fit: $\chi2/\text{df} = 3.345$; CFI = 0.916; TLI = 0.918; IFI = 0.925; RMSEA = 0.053. The results affirmed that hypotheses H1a, H2a, H3a, H4b, H4a, and H5 were supported, and H1b, H2b, and H3b were not supported. Table 7 summarises the path analysis results from the structural model.

**Table 7.** The path analysis results from the structural model.

| Relationships | Path Coefficients | Significant (P) Values | Test Result |
|---|---|---|---|
| H1a: Built environment capital → COVID-19 pandemic indices | **0.312** | *** | Supported |
| H2a: Disaster management indices → COVID-19 pandemic indices | **0.413** | ** | Supported |
| H3a: Awareness of COVID-19 → COVID-19 pandemic indices | **0.567** | ** | Supported |
| H4a: Built environment adaptive strategies → COVID-19 pandemic indices | **0.435** | *** | Supported |
| H1b: Built environment capital → Human social capital | 0.003 | 0.130 | Not Supported |
| H2b: Disaster management indices → Human social capital | 0.004 | 0.121 | Not Supported |
| H3b: Awareness of the COVID-19 pandemic → Human social capital | −0.406 | 0.526 | Not Supported |
| H4b: Built environment adaptive strategies → Human social capital | **0.359** | ** | Supported |
| H5: Human social capital → COVID-19 pandemic indices | **0.641** | *** | Supported |

Note: *** < 0.001, ** < 0.01.

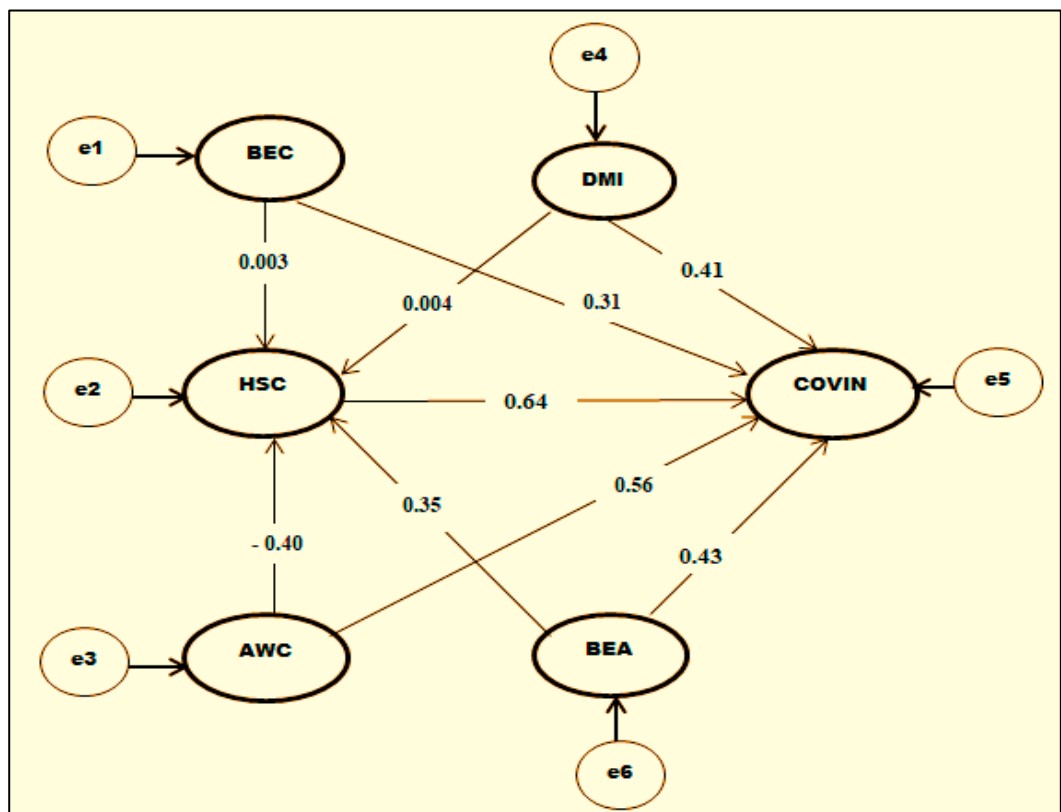

**Figure 8.** An overview of all hypothesised structural model results (BEC-built environment capital, DMI-disaster management indices, AWC-awareness of the COVID-19 pandemic, BEAS-built environment adaptive strategies, COVIN-COVID-19 pandemic indices, HSC-human social capital). Author's analysis.

In view of the results, built environment capital (BEC), disaster management indices (DMI), awareness of the COVID-19 pandemic (AWC), and built environment adaptive strategies (BEA) positively affect the COVID-19 pandemic indices (COVIN). Awareness of the COVID-19 pandemic (AWC) has the highest impact of 0.567 on COVID-19 pandemic indices (COVIN), followed by built environment adaptive strategies (BEC) with an impact of 0.435. Disaster management indices (DMI) have an impact of 0.413, while built environment capital (BEC) has the lowest impact of 0.312. Meanwhile, only human social capital (HSC) has a positive direct influence on COVID-19 pandemic indices (COVIN) with a path coefficient of 0.641. Thus, the hypothesis of built environment adaptive strategies (BEA) influence on human social capital (HSC) was supported with an impact of 0.359. The hypotheses of the impacts of built environment capital (BEC), disaster management indices (DMI), and awareness of the COVID-19 pandemic (AWC) on human social capital (HSC) were not supported by a significant value $p > 0.05$.

Table 8 shows that four constructs (built environment capital, disaster management indices, awareness of COVID-19, and built environment adaptive strategies) indirectly affect COVID-19 pandemic indices. The indirect effect of built environment capital on COVID-19 pandemic indices is 0.086, disaster management indices are 0.043, awareness of the COVID-19 pandemic is 0.144, and built environment adaptive strategies are 0.042. Overall, the total effect of human social capital on COVID-19 pandemic indices is higher than the others, with an effect of 0.753.

**Table 8.** The total effect coefficients.

| Path | Indirect Effect | Direct Effect | Total Effect |
|---|---|---|---|
| Built environment capital → COVID-19 pandemic indices | **0.086** | 0.000 | 0.086 |
| Disaster management indices → COVID-19 pandemic indices | **0.043** | 0.000 | 0.043 |
| Awareness of COVID-19 pandemic → COVID-19 pandemic indices | **0.144** | 0.000 | 0.014 |
| Built environment adaptive strategies → COVID-19 pandemic indices | **0.042** | 0.500 | 0.542 |
| Human social capital → COVID-19 pandemic indices | 0.000 | 0.753 | **0.753** |
| Built environment capital → Human social capital | 0.000 | 0.503 | **0.503** |
| Disaster management indices → Human social capital | 0.000 | 0.504 | **0.504** |
| Awareness of COVID-19 pandemic → Human social capital | 0.000 | 0.546 | **0.546** |
| Built environment adaptive strategies → Human social capital | 0.000 | 0.512 | **0.512** |

## 5. Discussion

### 5.1. Built Environment Capital and Human Social Capital in the COVID-19 Pandemic

This study raises our knowledge and comprehension of how diverse interconnections exist between the facets of a resilient built environment, which include built environment capital, disaster management, awareness of the COVID-19 pandemic, and built environment adaptive strategies, and how these, directly and indirectly, impacted the COVID-19 pandemic post-recovery. Social capital contributes to increased resilience during and after the COVID-19 pandemic. Most importantly, the outcome of our analysis revealed that human social capital had the highest impact on the COVID-19 pandemic indices. The results inferred that personal capacity to tackle the pandemic was reported as strong in all neighbourhoods, as corroborated by [71,72].

This study recognises the significance of resilience in the built environment and is in accord with various studies in urban planning, architecture, and disaster management [42,73,74]. This research has shown that strong social ties can contribute to effective disaster response, community support, and recovery. This finding aligns with current literature on the function of social capital in building community resilience (e.g., Putnam's theory of social capital). The research highlights that human social capital had the highest impact on COVID-19 pandemic indices. This finding suggests that individuals' connections, cooperation, and shared resources were pivotal in responding to and recovering from the pandemic. The effect of human social capital resonates with research on the importance of community engagement and social cohesion in times of crisis.

### 5.2. The Disaster Management Indices, Human Social Capital, and Resilient Built Environment Framework

Our results agree with a growing amount of empirical and theoretical research [30,75,76], indicating that people who live in close-knit communities recovered better during the outbreak for three reasons. Firstly, a deeper feeling of shared conviction exists in a more coherent community [77,78]. Secondly, when groups have high social capital, they are more inclined to organise themselves to provide communal assistance and encouragement for those particularly affected by a disaster, boosting or substituting state intervention [79]. Thirdly, in cultures with strong social capital, coping resources are more broadly available [30,80,81]. According to [80], social capital is a community-level interpersonal asset that helps to reduce the effects of disease, pandemic confinement, and seclusion.

### 5.3. Awareness of the COVID-19 Pandemic and its Connection to Human Social Capital

This study has affirmed that communities with strong social capital fare better during and after the pandemic in several ways, including increased compliance with public health guidelines, greater resilience to economic disruption, and better psychological well-being. These communities have been able to leverage their social networks and relationships

to provide mutual support, coordinate collective action, and, in the following situations, develop resilience amid disaster.

(i)   Mutual aid networks: In many communities, mutual aid networks have emerged during the pandemic to support vulnerable populations. These networks are typically composed of volunteers who offer to help with tasks such as grocery shopping, medication delivery, and transportation. These social networks coordinate their efforts to reach those in need. Communities with high social cohesiveness have been able to mobilise these networks with greater effectiveness and ensure that support reaches those who need it most; this accords with the studies of [78,81].

(ii)  Compliance with public health guidelines: Communities with strong social cohesion have generally been more compliant with public safety measures such as mask use, physical separation, and proper hand washing. This is because individuals in these communities are more likely to trust and respect each other, and wellness and good health are important to their fellow residents. This has contributed to lower rates of COVID-19 pandemic transmission in these communities, as supported by [2,82].

(iii) Resilience to economic disruption: Communities with strong social cohesion have been more resilient to economic disruption caused by the pandemic. This is because individuals in these communities are more likely to support local businesses and each other during economic hardship, in line with the studies of [73,83].

(iv)  Psychological well-being: Communities with strong social cohesion have also fared better in terms of psychological well-being during the pandemic. This is because individuals in these communities have a greater sense of social support, belonging, and connection to others [78].

*5.4. Adaptive Strategies for Mitigating the Impact of the COVID-19 Pandemic*

This study has revealed the influence of built environment adaptive strategies through intentional adjustments and modifications to the physical environment (such as buildings, public spaces, transportation systems, and infrastructure) to address and respond to changing circumstances, challenges, or threats. This is consistent with the study of [1,31,84]. Adaptive methods within the COVID-19 pandemic include changes to urban design, transportation systems, workplace layouts, public spaces, and housing to promote safety, hygiene, and social distancing. Additionally, communities with strong social networks, such as immigrant communities and religious groups, may have helped one another during the pandemic by providing financial assistance, food, and other essential resources. Positive actions could be taken through stakeholder participation in local community initiatives to tackle the COVID-19 pandemic threat. Social capital substantially influences people's resilience and communities' capacity to adapt to the COVID-19 pandemic's effects, as was corroborated by the discoveries of [31,77,84].

*5.5. Built Environment Capital, Disaster Management Indices, and COVID-19*
*Pandemic Awareness*

This study confirms that beneficial human activities can reduce the vulnerability of the built environment during the COVID-19 pandemic. Adapting to the COVID-19 pandemic will be simple in civilisations with strong community social connectedness [31,78,79]. That is consistent with prior research by [80–82] which established that the negative health effects of quarantine during a pandemic are reduced by social capital, an asset that exists at the neighbourhood level. In connection with previous findings, this study has shown that disaster response can improve connectedness among residents, and such efforts can increase self-confidence and shared understanding [83–85]. A proactive, resilient society or community acknowledges the certainty of change and works to build a system that responds to adaptable management procedures [86–89]. In terms of social cohesion and in connection with past studies, this study has shown that communities with high social cohesion, such as small towns and tight-knit neighbourhoods, have had lower rates of COVID-19 pandemic transmission compared to neighbourhoods with inadequate social capital.

## 6. Conclusions

The impact of a resilient built environment on post-COVID-19 recovery in Nigeria, employing human social capital as an intermediary factor, has been established. The results of this study indicate that built environment capital, disaster management indices, and COVID-19 pandemic awareness have indirect effects on COVID-19 pandemic indices through human social capital. The research offers valuable perspectives by revealing the significance of human social capital in enhancing adequate resilience in the built environment. This study affirms the significance of multi-disciplinary dimensions to enhance the resilience of the built environment, considering the intricate relationships among various factors, including social capital. It makes a substantial contribution to expanding knowledge in this area by uncovering multiple ways in which individuals, communities, and organisations are heavily reliant on social capital to recover from the adverse effects of the COVID-19 pandemic.

Notably, social capital often leads to effective responses compared to institutional assistance. While discussions about resilience in the built environment usually focus on post-disaster recovery procedures, this study underscores the ongoing importance of social capital for recovery following a disaster. It highlights how neighbourhoods rich in social connections tend to exhibit greater resilience during recovery phases. In the context of the built environment, resilience can include measures to minimise the transmission of infections, ensure access to essential services, and support economic recovery.

Achieving resilience in the built environment during and after the pandemic requires a multi-disciplinary approach involving professionals from various fields. In view of this, enhancing the resilience of cities against pandemics requires a multifaceted approach that involves urban planning, design, and infrastructure improvements. This study suggests some constructive urban design solutions, namely:

(i)     Resilient building design: The COVID-19 pandemic has underscored the critical role that resilient building design plays in ensuring the safety and well-being of communities. It is not only a response to the current crisis but also an investment in the future resilience of our cities. By promoting resilient building design through updated regulations, incentives, education, and collaboration, we can create a built environment that is better equipped to withstand and respond to future pandemics and health crises, ultimately enhancing the quality of life for all.

(ii)    Mixed-use zoning: Encourage mixed-use zoning to reduce the need for long commutes and create more walkable communities. This reduces pollution and helps in disease containment by reducing the need for extensive travel.

(iii)   Ensure critical infrastructure (hospitals, power stations, and water treatment plants) is designed to be adaptive and can continue functioning during and after pandemics.

(iv)    Green infrastructure: Integrate more green spaces, parks, and urban forests into the cityscape. These not only enhance the quality of life but also serve as natural buffers against disasters by absorbing excess water, reducing heat, and providing habitat for biodiversity.

(v)     Smart technology: Implement smart city technology for early warning systems and real-time monitoring of environmental conditions. This can aid in early detection and response to pandemics.

(vi)    Community engagement: Involve communities in the planning process. Engage residents in disaster preparedness and response planning to create a sense of ownership and resilience at the grassroots level.

(vii)   Disaster-resilient transportation routes: Design transportation routes, such as evacuation routes, to be more resilient to natural disasters, ensuring safe and efficient movement of people in times of crisis. Efficient and accessible public transportation systems reduce congestion, pollution, and the spread of diseases.

(viii)  Urban resilience education: Educational programmes are important to raise awareness and educate the public about disaster preparedness, response, and mitigation.

The limitations of this study include challenges in measuring social capital. A universally recognised approach for gauging social capital is lacking, and different approaches may yield different results. This poses challenges when assessing the influence of social capital strategies on resilience. Comprehending the relationships between adaptations in the built environment and human social capital during the pandemic has practical implications for urban planning, policy development, and public health interventions. These must prioritise the needs of vulnerable populations and ensure they receive the assistance they need to build social capital. This study's findings reveal insights for policymakers and other stakeholders in Nigeria and demonstrate the significance of considering human social capital in creating and strategising a resilient environment.

Policymakers can promote human social capital by investing in community-based programmes and initiatives that foster social capital and build trust among community members. This can include supporting community organisations, promoting the utilisation of communal areas for social interaction, and encouraging the development of neighbourhood-based networks and partnerships. It is important to take a holistic and inclusive approach to social capital-building in the built environment. This can involve engaging with diverse stakeholders and communities, identifying and addressing existing inequalities and power structures, and adopting a flexible and adaptive approach to resilience-building that acknowledges the intricate nature of the obstacles encountered by societies.

This study's proposed framework has the potential to function as a valuable instrument for policymakers and practitioners as they strive to bolster the resilience of constructed spaces against upcoming pandemics and calamities. This includes using materials and technologies resistant to natural disasters and other threats, as well as designing buildings and public spaces that are flexible and adaptable to changing needs and circumstances. Importantly, policymakers must also prioritise equitable access to resilient infrastructure and services. This means considering the needs of vulnerable populations, including but not limited to underserved communities, senior citizens, and individuals with limited abilities, and ensuring that they have access to safe and resilient buildings, public spaces, and services. There is a strong need for a multi-disciplinary strategy to enhance resilience in the built environment during the pandemic. Future research might include investigating the long-term impacts of social capital on resilience and exploring the potential for social capital to address other environmental challenges beyond the COVID-19 pandemic.

**Author Contributions:** Conceptualisation, O.P.A. and H.A.N.; methodology, O.P.A.; software, O.P.A.; validation, O.P.A. and H.A.N.; writing—original draft preparation, O.P.A. and H.A.N.; writing—editing, H.A.N. and Y.A.D.; visualisation, Y.A.D. All authors have read and agreed to the published version of the manuscript.

**Funding:** This research received no external funding.

**Data Availability Statement:** The data presented in this study are available on request from the corresponding author.

**Conflicts of Interest:** The authors declare no conflict of interest.

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
