# Peer review of "Strengthening Resilient Built Environments through Human Social Capital: A Path to Post-COVID-19 Recovery"

_urbansci, doi:10.3390/urbansci7040114_

Round 1

Reviewer 1 Report

It is an interesting and current topic, given the natural disasters occurring in various locations around the world.

It is an interesting work with a theme that highlights how social and constructive organization influences during times of disasters and pandemics. It delves into neighborly relationships or neighborhood organizations and urbanism or the built environment.

There are some aspects to improve:

- The hypotheses presented in the article (page 8) should be simplified for a better understanding of the study.

- Some images have low quality, for example, Figure 3, 4 or figure 8. It is recommended to improve the quality.

- The figures depicting data (Figures 1 and 2) do not have the same format, colors, text formatting. It is recommended to standardize the format used.

- I believe it would be appropriate to provide some constructive or urban design solutions to enhance the resilience of cities against natural disasters or pandemics.

Author Response

'Please see the attachment'

Reviewer 2 Report

In this manuscript, the authors obtained the basic information of the surveyed population using a questionnaire while proposing a framework based on PMT improvement, which contains various index analyses. The constructed model was analyzed through AMOS software to analyze the impact of built environment capital, disaster management index, COVID-19 pandemic awareness, and built environment adaptation strategies on the COVID-19 pandemic index and human social capital index. However, I suggest that the manuscript needs the following improvements before it is accepted for publication:

1. Firstly, the language of the manuscript is more problematic, and it is recommended to check the whole text for grammatical errors. Secondly. The manuscript has a very large number of low-level formatting errors; for example, there are colons and commas separating punctuation marks in keywords, and such cases should be consistent in context. For example, the first occurrence of an abbreviation of a noun should be given in full, and then it is sufficient to use the abbreviation in the following text. Still, there are many instances in the manuscript where it is abbreviated, many instances where it is in full, and some instances where the abbreviation is not given in full. The caption case for figures and tables is contextually consistent: all first letters are capitalized, or the first letters of all words are capitalized. Nouns are not contextually consistent, e.g. "COVID-19", but sometimes "Covid-19"; sometimes "Covid-19 pandemic", but sometimes "Covid-19 pandemic"; sometimes "Covid-19", but sometimes "Covid-19 pandemic". sometimes "Covid-19 pandemic", sometimes "Covid-19 epidemic". Please note that the manuscript contains more than these low-level formatting errors; please refer to the annotated PDF version for details.

2. Figures 1, 2, 3, and 4 are labeled with sources, but the remaining figures in the manuscript are not marked with sources. Meanwhile, Figure 4 is suggested to be re-generated as it is very unclear.

3. Table 1, Table 2, and especially Table 4 look very confusing, and it is suggested to redesign them. Columns 1 and 2 of Table 3 have poor correspondence of information and are very unintuitive, present redesign. In Table 5, it is recommended to use more abbreviations to improve readability. In Table 6, where there is no data, it is recommended to use uniform symbols to fill in to enhance simplicity and readability. The last three columns in Table 8 are not aligned; please optimize the table.

4. Lines 461 to 467 in the manuscript are less logical regarding the analysis of parameters; it is recommended to reorganize the language.

5. Lines 533 to 543 in the manuscript are about analyzing some indices. It is suggested to streamline the language and reorganize the logic.

6. The corresponding data of the five factors or how the data are classified, the author did not introduce, only gave the analysis. The sample of questionnaires is only 500; is the number of questionnaires sufficient? Is it guaranteed to be distributed evenly in the study area? Because uneven distribution may lead to biased results. It is suggested that the authors should increase the scope of the survey and the sample size to reduce the bias and give the analysis results after the detailed introduction of the conducted data. The 4th and 5th suggestions should be analyzed and discussed with the data presentation.

7. The reference citation format needs to be unified.

I suggest that the authors need to improve the quality of the English grammar in this manuscript.

Author Response

'Please see the attachment' 

Round 2

Reviewer 2 Report

In this manuscript, the authors obtained the basic information of the surveyed population using a questionnaire while proposing a framework based on PMT improvement, which contains various index analyses. The constructed model was analyzed through AMOS software to analyze the impact of built environment capital, disaster management index, COVID-19 pandemic awareness, and built environment adaptation strategies on the COVID-19 pandemic index and human social capital index. After a major revision, it can be noticed that the authors have addressed all the issues raised in the previous review and given a revision note for each of them. 

The quality of the revised English language has improved considerably.